# Motor Capacities in Boys with High Functioning Autism: Which Evaluations to Choose?

**DOI:** 10.3390/jcm8101521

**Published:** 2019-09-21

**Authors:** Véronique-Aurélie BRICOUT, Marion PACE, Léa DUMORTIER, Sahal MIGANEH, Yohan MAHISTRE, Michel GUINOT

**Affiliations:** 1UM Sports et pathologies, CHU Sud, CS 90338; avenue de Kimberley, F-38434 Echirolles CEDEX, France; MGuinot@chu-grenoble.fr; 2UF Recherches, CHU Grenoble Alpes, avenue de Kimberley, F-38434 Echirolles CEDEX, France; LDumortier@chu-grenoble.fr; 3INSERM U1042, HP² CHU Grenoble Alpes, avenue de Kimberley, F-38434 Echirolles CEDEX, France; marion.pace01@gmail.com (M.P.); inamiganeh@gmail.com (S.M.); yohan.mahistre974@orange.fr (Y.M.)

**Keywords:** children, autism spectrum disorders, motor impairments, cluster analysis

## Abstract

The difficulties with motor skills in children with autism spectrum disorders (ASD) has become a major focus of interest. Our objectives were to provide an overall profile of motor capacities in children with ASD compared to neurotypically developed children through specific tests, and to identify which motor tests best discriminate children with or without ASD. Twenty-two male children with ASD (ASD—10.7 ± 1.3 years) and twenty controls (CONT—10.0 ± 1.6 years) completed an evaluation with 42 motor tests from European Physical Fitness Test Battery (EUROFIT), the Physical and Neurological Exam for Subtle Signs (PANESS) and the Movement Assessment Battery for Children ( M-ABC). However, it was challenging to design a single global classifier to integrate all these features for effective classification due to the issue of small sample size. To this end, we proposed a hierarchical ensemble classification method to combine multilevel classifiers by gradually integrating a large number of features from different motor assessments. In the ASD group, flexibility, explosive power and strength scores (*p* < 0.01) were significantly lower compared to the control group. Our results also showed significant difficulties in children with ASD for dexterity and ball skills (*p* < 0.001). The principal component analysis and agglomerative hierarchical cluster analysis allowed for the classification of children based on motor tests, correctly distinguishing clusters between children with and without motor impairments.

## 1. Introduction

Autism spectrum disorders (ASD) are a range of signs classified as neurodevelopmental disorders that are characterized by social and communication deficits with abnormal repetitive and stereotypic behaviors [1]. Children with ASD also show significant delays in motor development, impaired movement performance and impaired motor planning compared to their typically developed peers [2,3]. More attention has been given to people presenting with motor skill difficulties as a result of autism literature, and several studies have shown particular features in early motor development among children with ASD. Motor deficits were once seen as being relatively frequent in children with ASD and there is a growing consensus on the idea that motor development might be atypical depending on the individual [4,5,6,7]. Thus, the motor movement of these children is often described as clumsy, slower and less fluent, with stereotyped behavior and impaired gross motor skills [4,5,6,7]. Given the emerging findings of early motor impairment in young children with an ASD [4,5,6,7], understanding the relationship between gross motor capacities and later physical fitness in these children is of particular interest.

Specifically, when compared to typically developed children, children with ASD are more likely to have difficulties with balance, postural stability, gait, joint flexibility, and movement speed [6,8,9,10,11,12,13]. Some children with ASD have general dynamic coordination disorders including locomotion, jumping, and dynamic balance [13]. This coordination is essential during every physical activity that requires strength (hitting or throwing a ball, walking on a line, balancing). Activities must include combinations of asymmetrical or symmetrical adjustments involving bilateral control, dissociation or an association of regulations between upper and lower limbs [12]. Thus, these deficits in motor coordination and postural stability in individuals with ASD have been confirmed through a meta- analysis [3]. These authors have concluded that balance training early in development might help to prevent the subsequent emergence of deficits in other motor abilities. Nevertheless, given the evidence of this atypical development in autism, it is also important to examine the potential relationships between early locomotor activity and the physical fitness of these children. The benefits of sports participation for physical and mental health are widely recognized [14]. Lack of physical activity is regarded as a life course risk factor for cardiovascular disease, diabetes, cancer, and obesity [15]. Conversely, meeting guidelines for physical activity for children—defined as 60 min of moderate-to-vigorous intensity daily physical activity—is associated with beneficial health outcomes across a variety of physical and mental conditions [16]. In addition to the positive influence on the child’s general physical profile, involvement in sport is also associated with the development of sport-specific characteristics and contributes to the development of better cardiorespiratory fitness [17,18,19]. Therefore, it seems essential to stimulate children with ASD through a wide range of physical activities. Evaluating all motor skills is important in order to properly identify the specific needs of these children.

In the present study, we investigated the motor capacities of children with ASD compared to typically developed children in order to provide a physical fitness profile of this population. We used three different tests that are related to specific aspects of motor activities and we assessed cardiorespiratory fitness with an evaluation of maximum aerobic capacity (VO_2max_). According to the American College of Sports Medicine, health-related physical fitness includes at least body composition, aerobic capacity (VO_2_), muscular strength, and flexibility [20].

The second aim of this study was to identify which motor tests best discriminate children with or without ASD. 

## 2. Methods

### 2.1. Participants 

The study was approved by the Ethics Committee of the Hospital of Grenoble (France) (N°A00-865 40) and was registered on the clinicaltrials.gov registry with the number NCT 02 830 022 (12 July 2016). All experiments were performed in accordance with relevant guidelines and regulations. Each parent and child received information about the purpose and nature of the protocol. Consent was obtained from the child and from his parent and/or legal guardian.

Forty-two young male children, coming from ordinary schools or local support groups, were recruited and participated (Table 1). Twenty were typically developed children and served as controls (CONT) and twenty-two were children with ASD (ASD). The diagnosis of ASD was confirmed by experienced physicians and psychologists according to the Diagnostic and Statistical manual of Mental Disorders-V DSM-V [1]. Children with ASD were also assessed with the Autism Diagnostic Observation Schedule (ADOS) [21] and this diagnosis, along with the ADOS, was an inclusion criterion. Intellectual quotient (IQ) was assessed using the Wechsler Intelligence Scale for Children, 4th Edition [22]. The inclusion IQ criterion concerned children with an IQ >70 (children with intellectual disabilities (IQ < 70) were not included). In line with ethical guidelines, IQ scores and ADOS results were not made available to researchers. Nevertheless, based on the results from a clinical and competent psychologist diagnosing children with ASD, IQ scores were certified as being >70, and diagnoses of autism were confirmed for all children with ASD included in this study. Additionally, the Vineland scale was administered to every child during their first medical visit [23]. This scale is used to evaluate three dimensions of daily adaptive behaviors (communication, daily living skills and socialization). Children (CONT or with ASD) presenting with cardiac or respiratory disease (that might alter heart rate (HR) response) and those taking medication or with co-morbid medical or psychiatric disorders were not included in the study.

Each child underwent a medical examination and a resting 12-lead electrocardiogram after five minutes in supine position (HR_rest_) to confirm the absence of contra-indication before performing the motor assessment. Body mass index was calculated using the following equation: BMI = body weight (kg)/height^2^ (m^2^). The pubertal stage of children was assessed using the Tanner criteria [24]. Aerobic capacity (VO_2peak_ measure) was assessed during a maximal exercise treadmill test. Each participant underwent a continuous maximal incremental protocol, walking until exhaustion, following the protocol described elsewhere [11]. All CONT children as well as 21 out of 22 children with ASD completed this cardiorespiratory assessment. One case failed to obtain results due to the non-cooperation of a child who refused the application of electrodes from the electrocardiogram, a compulsory step in the process of the cardiorespiratory test. However, this child performed all the other motor tests without any difficulty. The characteristics of the two groups were recorded in the Table 1.

### 2.2. Motor Assessment

According to the American College of Sports Medicine, health-related physical fitness includes body composition, aerobic capacity, muscular strength, and flexibility [20]. Although it is not directly related to health and wellness, skill-related fitness includes the components of agility, balance, coordination, speed, power, and reaction time [25]. All fitness tests in the study were carried out by an experienced exercise physiologist. Children performed all tests in standardized conditions (health and sanitary requirements, adequately dimensioned space, lighting, adequate flooring, sufficient equipment, accessories and measuring instruments) [17,26,27]. Demonstrations, verbal explanations and/or pictograms were in some cases used to help children to better understand the tasks. Each child was individually tested by the same experienced therapist. There was no precise order for completion of the motor tests as they were independent from each other. 

A maximal exercise treadmill test was conducted to evaluate aerobic capacity (methodological details and data published elsewhere [17,26,27]).

Motor assessment of the children was performed using three different tools: the Movement Assessment Battery for Children (M-ABC; [26].), the Physical and Neurological Exam for Subtle Signs (PANESS; [27]) and the European Physical Fitness Test Battery EUROFIT [28].

Of all standard motor tests, the M-ABC is the most commonly used. It is a norm referenced test designed to identify motor difficulties in children aged from 4 to 12. The test includes eight items grouped into three dimensions (manual dexterity, ball skills, and balance). The sum of the eight-item scores comprises the total impairment score of the tests and ranges between 0 and 40; a lower score represents a better movement execution outcome. In the present study, boys aged 7 and 8 were tested on Age Band 2, those between 9 and 10 were tested on Age Band 3 and those between 11 and 12 were tested on Age Band 4 (in reference to the classification of Henderson and Sugden [26] (for details, see Appendix A)).

The PANESS battery is a standard childhood assessment that evaluates several different categories of motor tasks, including stressed gaits, balance, repetitive timed movements, patterned timed movements, and the presence of subtle neurological signs, such as overflow movements, abnormal posturing, and dysrhythmia. This battery includes a total of 21 tests, and the higher the value, the lower the performance (for details, see Appendix A).

The EUROFIT battery is designed for the assessment of health-related fitness in children and adults [28]. The test is a set of different physical fitness tests covering balance, flexibility, speed, endurance and strength (see Appendix A). Seven items of this test battery were completed by the children. The standardized and validated test battery was designed by the Council of Europe for children of school age and has been used in many European schools since 1988. Two other additional tests (reactive speed and motor educational course) classically used in French schools were proposed. 

### 2.3. Statistical Analysis

Results are given as mean ± standard deviation (SD). First, to compare motor test data differences between the ASD and CONT groups, a Mann–Whitney test was performed (Statistica Software 8.0). Significance was accepted when *p* < 0.05. 

The second objective was to identify which motor tests best discriminate children with or without ASD. For this purpose, a principal component analysis (PCA) and an agglomerative hierarchical cluster analysis (AHCA) were performed (R software^®^ [29]). These methods allowed us to highlight the variables that best discriminated the two groups (ASD/CONT) [30]. However, some preliminary steps were necessary.

In this study, we had *n* = 42 observations (20 CONT/22 ASD) and *p* = 47 predictors (anthropometric characteristic and motor test results). The principal components were supplied with the normalized version of the original predictors. Here, normalization and centralization of the data by the feature scaling method was first applied. It consisted of determining the difference between the predictor and its minimal, divided by the difference between the maximum of the predictor and its minimum to reduce the scale of our predictors to (0–1). The following formular (Formula (1) was used to achieve this: (1)$$\frac{X-\min\left(X\right)}{\max\left(X\right)-\min\left(X\right)}$$
After this first step, we used the elastic net method which simultaneously permits automatic selection of pertinent variables. It can also select groups of correlated variables [31]. With this elastic net method, we obtained a selection of *p* = 8 pertinent variables. The ROC (Receiver Operating characteristic) curves showed superior performance with the elastic net method (Area Under the Curve (AUC) = 0.91) compared to other feature selection methods (Figure 1; AUC = 0.67 with all variables; AUC = 0.86 with a selection of 20 discriminant variables selected with methods based on the Wilcoxon test and the Benjamini–Hochberg test). 

Finally, PCA and AHCA were performed to obtain the classification of individuals using a motor profile (only on the eight pertinent variables) selected with the elastic net method. A dendrogram and factor map were proposed. These graphical representations indicate the height of the fusion provided on the vertical axis (the higher the height of the fusion, the less similar the clusters are) in the dendrogram, and the partition of our data into clusters on the factor map. 

## 3. Results

Participant characteristics are shown in Table 1. There were no statistical differences in the anthropometric characteristics between the two groups (Table 1), but Vineland results confirmed the significant alterations of adaptive behavior in the ASD group, which obtained lower scores than control children in each dimension (Table 1). Children with ASD showed motor impairments relative to the controls (Table 2 and Table 3). Specifically, flexibility (*p* < 0.05), explosive power (Broad Jump, *p* < 0.01) and strength scores (hand grip: *p* < 0.01 and sit up tests: *p* < 0.05) were significantly lower in ASD compared to control children (Table 2). The ASD group presented a lower VO_2peak_ than the controls (*p* < 0.01; Table 2). Children with ASD also needed a significantly longer time to achieve the motor educational course (*p* < 0.01; Table 2).

Total score on the M-ABC showed a weaker performance from children with ASD compared to the CONT group (*p* < 0.001; Table 2) with dexterity and ball skill scores significantly higher in the ASD group (*p* < 0.001). 

Moreover, children with ASD presented with motor skill impairments, with significantly higher PANESS scores than CONT children for gait and balance (*p* < 0.001), dysrhythmias (*p* < 0.001) and overflow (*p* < 0.05; Table 3). 

### 3.1. Principal Component Analysis

A PCA was made only based on the eight significant variables which helped explain 68.8% of the variance on the first two dimensions (47.6% and 21.2%), which was a satisfactory outcome. Figure 2 shows the results of the PCA biplot obtained by children on motor tests.

Two clusters were determined with the blue space representing the CONT group and the yellow space representing children with ASD. Four children with ASD [23,24,26,27] were misclassified and their scores appeared in the CONT group (blue cluster), and one CONT child (k) was classified outside of this cluster.

In this graphic representation, a child on the same side of a given variable obtained a high score for this variable. A low value for this variable was attributed to a child on the opposite side. It appeared that the CONT group tended to have higher values in muscular endurance (EUROFIT sit up test) and aerobic capacity (VO_2_) than the ASD group. These CONT children also made fewer mistakes on the dexterity tests (items B–C of the M-ABC). Conversely, the ASD group tended to have higher values on the other four tests (tandem walking backward; walks on side of feet; ball skills (item E); ball skills (item D)) for which the higher the score, the lower the motor capacity.

### 3.2. Agglomerative Hierarchical Cluster Analysis (AHCA)

An agglomerative hierarchical cluster analysis allowed for the classification of individuals based on the motor tests. This analysis ranks each child on a dendrogram and on a factor map, using different colors, in order to indicate the membership of specific items from different clusters.

On the factor map (Figure 3), four clusters were obtained that successfully discriminated children into: Cluster 1 (red) regrouped the controls and three children from the ASD group (subjects 23, 24 and 27). This cluster included children who had the best scores on muscular endurance and VO_2_ values, and low error scores on other motor items.Clusters 2 (green) and 4 (purple) only regrouped children with ASD. In these two clusters, these children have the lowest VO_2_ values and the highest error scores.Cluster 3 (blue) included two CONT and five ASD children. The two CONT children have the highest VO_2_ values (66.1 mLO_2_·kg^−1^·min^−1^ and 69.3 mLO_2_·kg^−1^·min^−1^, respectively) but they made many mistakes on the dexterity tests, balls skills and walk tests (seven and nine, respectively). The children with ASD in this cluster have the same profile: high VO_2_ values (mean = 53.2 mLO_2_·kg^−1^·min^−1^) but made many mistakes (mean = 10.8).

## 4. Discussion

The present study focused on the motor capacities of children with ASD and on motor tests to assess the motor capacities of children. Our results confirmed that children with ASD displayed poorer gross motor skills than CONT children even though individual variability was considerable in this population. Some motor impairments were well identified in the studies of children with ASD. They reported fine and gross motor coordination, muscle tone, strength, praxis, performance of repetitive movement, subtle neurological signs of overflow, dysrhythmia, and motor discontinuity [13,27,32,33,34]. The conclusions of these works were convergent and could be used to estimate the existence of motor deficits in children with ASD.

In this study, an exhaustive assessment of the motor profile by 42 motor tests confirmed the high frequency of motor impairments in the ASD population. According to our previous results [17], a lower aerobic capacity (VO_2_) in the ASD group compared to the CONT group was observed, alongside lower strength and flexibility (Table 2). In this previous study, children with ASD had specific lower VO_2peak_ compared to controls, with significantly impaired running performance during the maximal treadmill test (final slope, speed and effort duration). They also presented with high difficulties in balance, postural stability, movement speed and strength, and the authors reported new results showing a lower physical fitness in children with autism.

In the present work, lower scores were observed in gait and balance, dexterity, and ball skill items (Table 2 and Table 3). These observations were concordant with those obtained by others, who used either EUROFIT or PANESS or M-ABC [3,8,13,27,32,33,34,35,36,37,38,39,40,41,42,43,44,45]. Indeed, Dickinson and Place [34] reported similar results from 100 children with ASD that were evaluated with EUROFIT. They confirmed the adequate reliability of these motor tests with this population in order to control the effects of a computer-based activity program on physical fitness. Green et al. [6] measured movement skills in a large group of 101 children using the M-ABC and showed that 79% of children with ASD presented with movement impairments. At the same time, Jansiewicz et al. [37] used the PANESS and found that the ASD group had significant impairments on several measures of motor control compared to typically developed children. The same motor impairments were observed in this study. Manual dexterity, ball skills, gait and balance, repetitive or speed movements, and a motor educational course were tasks that required some physical capacities such as strength, flexibility, balance, coordination, and motor planning. In the ASD group, these motor capacities were impaired. The strength of children with ASD was weaker (results on muscular endurance and hand grip tests) and this might have played a role in other specific activities such as locomotion and reaching tasks [4]. Flexibility is a capacity that allows a large range of motion. Any limitation inevitably leads to an increase in muscle energy expenditure in order to compensate for this limitation. Flexibility is therefore an important element of physical fitness, and a lower performance on the flexibility task contributes to a lower physical fitness in ASD.

With the PANESS, and more specifically with the assessment of the gait and balance dimension, children with ASD presented with many difficulties. They made more mistakes on each of the five tests involved in walking on a line, and they had difficulties maintaining balance on one foot and hopping on one foot compared to CONT children. This observation was confirmed by the PCA analysis, which used these significant variables to discriminate the two groups of children. Some mechanisms were proposed to better understand the motor deficits observed in ASD, such as abnormalities in the cerebellum, a key element in sensorimotor control [46], or dysfunctions in frontal–striatal connections, which are involved in motor, cognitive, and behavioral functions within the brain [3]. Moreover, the PANESS made it possible to assess the quality of movement. Scores of dysrhythmias (*p* < 0.001), of impersistence (NS, but 3.5 times higher), and of overflow (*p* < 0.05) were higher in the ASD group. Several researchers suggested that children with ASD had dyspraxia when compared with a control group, and particularly that this dyspraxia may be linked to a delay in spatial mapping [47]. Glazebrook et al. [48] demonstrated that children with ASD are characterized by a slower preparation time to plan a movement and need more time to execute precise movements. Accordingly, our results on a motor educational course and PANESS tests confirmed these observations and the difficulty for children with ASD to efficiency and quickly plan a movement, as well as being performant in a timed test.

Our results therefore confirmed the motor deficits in children with ASD. The novelty in this work was the selection of relevant variables that made it possible to properly discriminate between the types of children according to their motor profile. Thus, it allowed us to obtain interesting information concerning the care of a child by evaluating them using the eight tests selected by the PCA without having to undergo a long evaluation on a large number of tests.

These results also confirmed the importance of motor stimulation at a very early age in order to develop a complete motor profile, adequate skills, and to increase their effectiveness. To develop the physical fitness of the child, a regular physical activities program combining components of aerobic, strength, flexibility and dexterity exercises, ball games and neuromuscular training for maximum gains in fitness and body composition is recommended.

The second aim of this study was to highlight the variables that best discriminated the children with or without ASD. With the PCA and AHCA methods, an interesting classification was obtained. These two methods were complementary, as they can both reproduce information to discriminate children, and were also beneficial to determine relationships between particular motor tests and children.

Firstly, we observed two distinct groups, one with CONT children and one with children with ASD (Figure 2). Secondly, we showed that the classification in the dendrogram and in the factor map was better (Figure 3). Four clusters were obtained as indicated on the factor map. Cluster 1 included 90% of CONT children, whereas Clusters 2, 3 and 4 were associated with 86.4% of the children with ASD with lower motor skill scores.

On this representation, three children with ASD (children 23, 24 and 27) were classified in Cluster 1, a cluster largely composed of CONT children. These children with ASD were those who obtained the best scores on the motor tests, with especially high values in aerobic capacity and strength tests, the essential determinants of physical fitness. Two CONT children (g and k) were misclassified in Cluster 3. These children had obtained very good results on the cardiorespiratory assessment but they had made many mistakes on the other tests. This profile confirms that good physical fitness is a set of motor skills and not just a single capacity.

In Figure 2, axis H and G (muscular endurance and VO_2_ tests) were the assessments that indicated which physical capacities were lacking the most in the ASD group. Axes A, B, C and D were the four axes that showed in which capacities CONT children had more superior performances than the ASD group.

Consequently, to establish a pertinent motor profile, it was essential to measure VO_2_ and strength, which were the skills that could best discriminate the two groups (and possibly the skills needed to complete the dexterity and ball skills tests (the PANESS, for example)).

Thus, even if the AHC analysis was not purposive in finding out certain relationships of interest, it objectively represented relationships between groups. The AHC analysis was much better at specifying exact relationships as the distance matrix provided a greater resolution than the PCA (Figure 2). Indeed, AHC analysis (Figure 3) classified two of the CONT children in a different group than the other CONT children (subjects g and k; Cluster 3) and three children with ASD in Cluster 1, which was mainly represented by CONT children. This relationship could not be deduced by using PCA. Otherwise, even though the analyses were different, the same conclusions were drawn from the data observed in the PCA and the AHC analysis: children with ASD were characterized by motor impairments.

Nevertheless, some limitations still need to be considered. Firstly, in order to increase statistical power and confidence in the generalizability of the results related to the classification of subjects, there is a need for higher recruitment of children. Secondly, because only boys with IQ >70 were included in this work, it will be necessary to reproduce the same study with children with more severe ASD. Finally, future research should also consider the use of samples with girls to allow for further subgrouping, examining older or younger children with ASD.

## 5. Conclusions

Although impaired motor activity is not included in the diagnosis of ASD, motor skill difficulties appeared to be an observable trend. Limitations in motor activity in children with ASD might reduce opportunities for social interactions and learning. To promote good physical fitness for children with ASD, it is therefore important to improve aerobic capacity and strength through a large variety of activities based on dexterity and ball skills. Moreover, the analysis of motor capacities using principal component and agglomerative hierarchical cluster analysis in children with ASD successfully identified distinct clusters. The originality of these statistical methods made the discrimination between children with ASD and typically developed children possible according to their motor difficulties and not only according to their diagnosis of autism alone. The motor profile obtained in this study using the eight tests identified could be used as a clinical evaluation because it provides pertinent information for the care of a child with ASD without having to subject them to a tedious evaluation based on numerous tests.

## Figures and Tables

**Figure 1 jcm-08-01521-f001:**
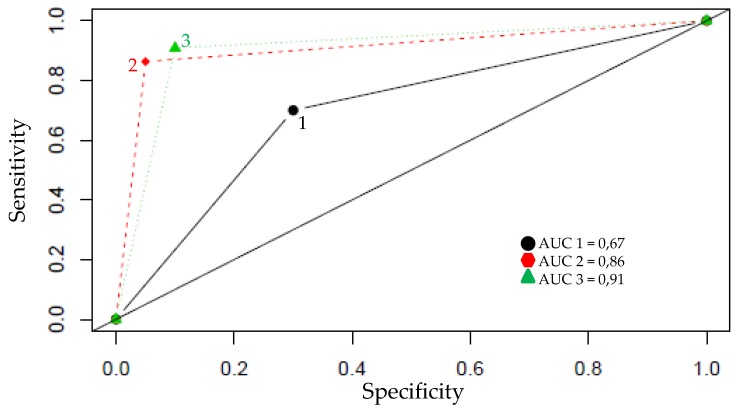
ROC curves and the area under the ROC curve (AUC).

**Figure 2 jcm-08-01521-f002:**
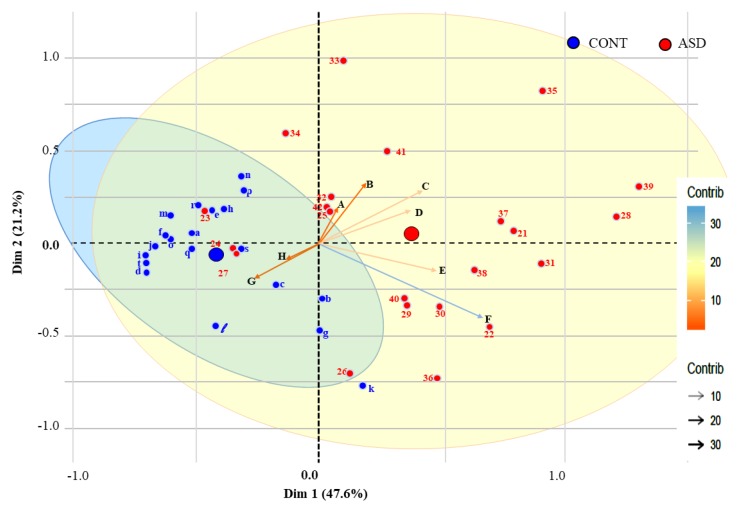
Principal component analysis (PCA) biplot. A: tandem walking backward; B: walks on side of feet; C: ball skills (item E); D: ball skills (item D); E: dexterity (item B) F: dexterity (item C); G: EUROFIT sit up test; H: VO_2_; Contrib = contribution; Dim: dimension. Children are represented from a to t for the CONT group (*n* = 20) and 21 to 42 for children with ASD (*n* = 22).

**Figure 3 jcm-08-01521-f003:**
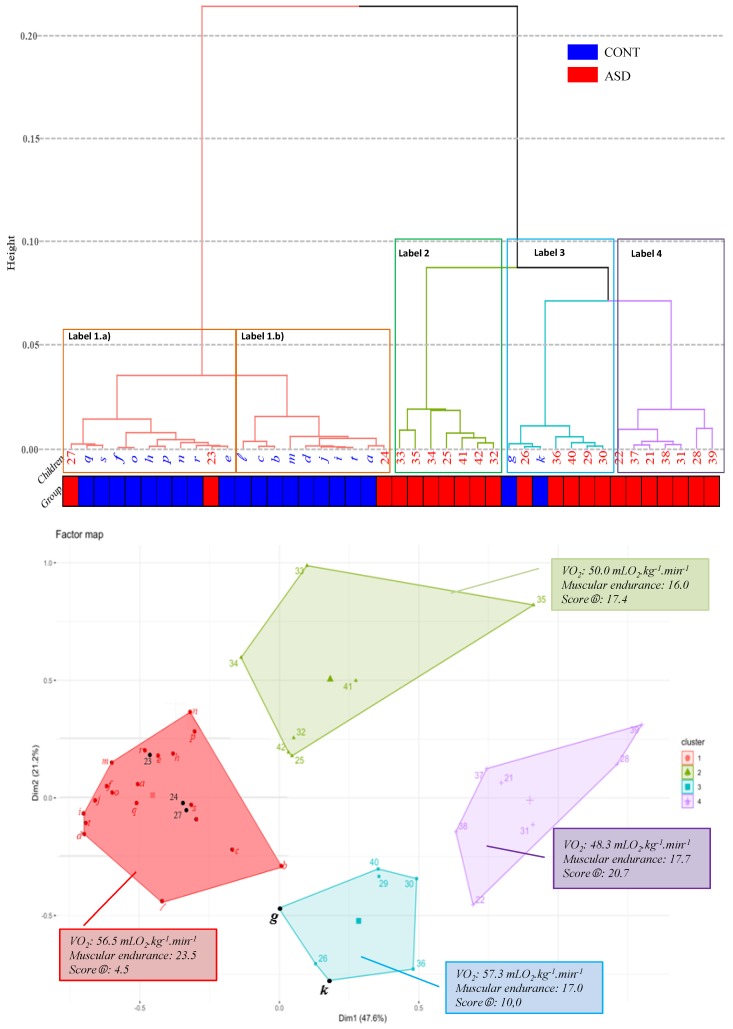
Dendrogram and factor map obtained by agglomerative hierarchical cluster (AHC) analysis. Children are represented from a to t for the CONT group and 21 to 42 for children with ASD. Score
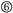
 = sum of errors on the 6 variables selected by the PCA (items B; C; D; E of M-ABC; tandem walking backward; walks on side of feet). Label 1: mainly characterized by CONT children (18 CONT and three ASD), with very good VO_2_ values (mean = 56.5 mLO_2_·kg^−1^·min^−1^) and with very few errors during motor evaluations. Two subgroups were observable depending on the muscular endurance: in label 1a, the mean value was lower than in label 1b (18.1 versus 29.6, respectively). Label 2: exclusively represented by children with ASD who had the same motor profile with average assessments. Label 3: mainly characterized by ASD children (five ASD + two CONT). In this group, children had satisfactory results, but the errors made on the tests were too numerous (>10) to be classified in another label. Label 4: exclusively represented by children with ASD who had the same motor profile, with low assessments.

**Table 1 jcm-08-01521-t001:** Participant’s anthropometric characteristics and Vineland assessment.

	CONT (*n* = 20)	ASD (*n* = 22)
Age (years, min–max)	10.0 ± 1.6 (8–12)	10.7 ± 1.3 (8–12)
Weight (kg)	33.3 ± 7.2	36.0 ± 13.3
Height (cm)	141.0 ± 10.5	144.7 ± 8.7
BMI (kg/m^2^)	16.0 ± 1.5	16.8 ± 3.8
Ratio waist/hip	0.88 ± 0.05	0.89 ± 0.05
Tanner stage 1 (no. of subjects)	16	16
Tanner stage 2 (no. of subjects)	4	6
**Vineland Assessment**		
Communication	121 ± 5	105 ± 12 ***
Daily living skills	132 ± 12	114 ± 13 ***
Socialization	106 ± 9	88 ± 11 ***

Values are means ± standard deviation (SD). CONT: controls; ASD: autism spectrum disorder; BMI: body mass index (see methods section). Tanner stage: number of subjects in this stage. Vineland assessment: the higher the score, the lower the deficit. Significantly different from CONT *** *p* < 0.001.

**Table 2 jcm-08-01521-t002:** Motor assessment by European Physical Fitness Test Battery (EUROFIT) and the Movement Assessment Battery for Children (M-ABC).

Aerobic Capacity	CONT	ASD
VO_2peak_ (mLO_2_·kg^−1^·min^−1^)	58.1 ± 8.8	50.6 ± 8.9 **
**EUROFIT and Additional Tests**		
Flamingo Balance Test (s)	29.4 ± 2.7	26.4 ± 6.6
Plate Tapping Test (s)	41.9 ± 9.7	43.4 ± 9.4
Sit and Reach Flexibility (cm)	−16.4 ± 8.3	−22.0 ± 7.1 *
Broad Jump (cm)	132.0 ± 22.6	113.8 ± 23.6 **
Vertical Jump (cm)	24.2 ± 5.9	21.4 ± 7.1
Hand Grip Strength Test (a.u)	162 ± 51	127 ± 39 **
EUROFIT Sit Up Test (n)	23.2 ± 6.3	17.5 ± 5.2 *
Reactive Speed (ds)	22.7 ± 3.3	21.0 ± 4.1
Motor Educational Course (s)	14.5 ± 2.3	18.9 ± 5.4 **
**M-ABC**		
Dexterity (score total of items A, B and C)	2.08 ± 3.02	7.22 ± 4.28 ***
Ball skills (score total of items D and E)	0.76 ± 1.22	3.41 ± 3.03 ***
Balance (score total of items F, G and H)	1.84 ± 2.00	3.39 ± 3.58
Total score of M-ABC	4.68 ± 3.95	14.02 ± 8.29 ***

Values are means ± SD. Significantly different from CONT * *p* < 0.05; ** *p* < 0.01; *** *p* < 0.001. Low negative values of flexibility represent a poorer performance. Strength: the higher the value, the higher the performance. M-ABC score: the higher the value, the lower the performance. VO_2peak_: peak aerobic capacity; s/ds: score in seconds or deciseconds; a.u: arbitrary unit; n: number of sit-ups.

**Table 3 jcm-08-01521-t003:** Motor assessment by the Physical and Neurological Exam for Subtle Signs (PANESS) battery.

PANESS	CONT	ASD
Lateral preference pattern (1)	1.75 ± 1.01 (12 Right/2 Left/5 mixed/1 eye alone)	1.90 ± 1.26 (14 Right/4 mixed/4 eye alone)
Walks on heels (2)	2.9 ± 1.6 (0 to 7)	3.5 ± 3.3 (1 to 15)
Walks on tiptoe (2)	0.2 ± 0.4 (0 to 1)	0.6 ± 0.8 (0 to 3) *
Walks on side of feet (2)	1.3 ± 1.3 (0 to 5)	3.1 ± 2.3 (0 to 8) **
Tandem walk forward (2)	0.3 ± 0.6 (0 to 2)	2.2 ± 3.6 (0 to 15) *
Tandem walk backward (2)	1.1 ± 1.1(0 to 4)	3.4 ± 3.5 (0 to 15) **
Sustentation posture (3)	18.5 ± 4.0	15.9 ± 6.5
Sustentation steadiness (3)	19.6 ± 1.6	19.9 ± 0.2
Finger to nose (L+R) (2)	0.1 ± 0.44 (0–2)	0.32 ± 0.72 (0–2)
Tongue protrusion (3)	20.0 ± 0.0	18.6 ± 3.7 NS
Stand on one foot (mean L+R) (3)	28.6 ± 3.4	24.3 ± 7.7 *
Hop on one foot (mean L+R) (4)	49.8 ± 0.9	41.9 ± 15.2 *
Foot tap (mean L+R) (3)	7.0 ± 1.8	7.0 ± 2.1
Foot heel toe tap (mean L+R) (3)	9.3 ± 2.5	11.0 ± 4.2
Hand pat (mean L+R) (3)	6.1 ± 2.1	6.0 ± 1.7
Hand pronation/supination (mean L+R) (3)	7.9 ± 2.0	8.1 ± 2.0
Finger tap (mean L+R) (3)	6.8 ± 1.1	6.3 ± 1.4
Finger succession (mean L+R) (3)	10.6 ± 2.8	11.0 ± 3.5
Tongue wiggles side to side (3)	9.0 ± 2.3	8.5 ± 2.6
**Global score of PANESS**		
Gait and balance total score	6.60 ± 3.64	17.50 ± 12.68 ***
Dysrhythmias	1.3 ± 1.2	3.0 ± 1.3 ***
Impersistence	0.60 ± 1.32	2.09 ± 3.68
Involuntary movement score	1.40 ± 1.14	1.40 ± 1.50
Repetitive speed of movement score	12.80 ± 3.06	15.22 ± 9.2
Overflow, grand total	5.75 ± 4.43	10.0 ± 7.26 *

(1) = laterality tests: the score is obtained by a code (all items Right = 1; all items Left = 2; some Right and Left = 3, Eye alone different from other = 4). (2) = number of errors. (3) = score in seconds (4) = number of hops. Significantly different from CONT * *p* < 0.05; ** *p* < 0.01; *** *p* < 0.001. Detailed procedures for scoring PANESS results are provided by Denckla et al. [27].

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
