# Peer review of "Motor Capacities in Boys with High Functioning Autism: Which Evaluations to Choose?"

_jcm, 2019, doi:10.3390/jcm8101521_

Round 1

Reviewer 1 Report

The authors studied the motor capacities in 22 boys with high functioning ASD compared to 20 normally developing children and found significant differences in a variety of parameters of the motor function between the groups. Such studies were also carried out by other investigators. However, the authors used many different parameters and a large variety of tests, a fact that adds strength to this study. The manuscript is well written, and the authors also delineate the weaknesses of their study (a small sample size, only high functioning ASD, only boys). Due to these weaknesses, the authors might prefer to be more careful in their conclusions (last sentence of the conclusion section) where they advise the use of several of the tests for the delineation of the motor dysfunction of children with ASD in order to better plan treatment. They should probably write that more children have to be tested in order to be able and advice the use of these tests for planning treatment. In addition, it might be advisable to change the title adding the fact that the study relates to ASD boys with high function. The published literature is well discussed and their results are clearly presented.

his is a well- written study evaluating the motor function of boys with high function ASD. The strength of the study is the use of many different tools for the evaluation of motor function, allowing the authors to suggest the proper tools for motor function measurement and even advice to use them for the planning of treatment. The weakness, however, is the relatively small sample.

Author Response

The authors might prefer to be more careful in their conclusions (last sentence of the conclusion section) where they advise the use of several of the tests for the delineation of the motor dysfunction of children with ASD in order to better plan treatment. The motor profile obtained in this study using the 8 tests identified could be used as a pertinent clinical evaluation because it provides pertinent information for the care of a child with ASD, without having him to go through a tedious evaluation based on numerous tests.

We proposed a more careful conclusion:

It might be advisable to change the title adding the fact that the study relates to ASD boys with high function.  

A new title was proposed: “Motor capacities in boys with high functioning autism: which evaluations to choose?”

Reviewer 2 Report

The manuscript describes a study investigating the relation between ASD and motor capacities of children. The study appears solid, the text is well condensed, and I appreciated reading the manuscript. Still, the investigated relation between ASD and motor capacities is nothing new and as the authors also report, a range of previous studies have identified similar relationships. I believe that the manuscript does not fully identify the knowledge gap that it is supposed to fill. Do not get me wrong, the study has many interesting properties in that it combines several methods for motor assessment were combined. I think there are several potentially important contributions hidden in this study, but currently not spelled out. Also, I miss a proper background section (the paper jumps directly from introduction to methods), but the absence of background is to a large degree compensated by a literature review included in the discussion.

The main finding highlighted is that flexibility, explosive power, strength, dexterity, and ball skills are lower in children diagnosed with ASD, compared to controls. Most of these results aligned with previous research, and here I would like to see a clearer presentation of what is new.

The authors claim that “These results also confirmed the importance of the motor stimulation at a very early age in order to develop a complete motor profile, adequate skills, and to increase their effectiveness.” In this regard I’m much more sceptical. Firstly, the age span is large, starting at the age of 4, and it’s difficult to assess anything about children in a “very early age” based on this data. Secondly, and even more problematic, is that the study demonstrates a correlation between motor skills and ASD (not a casual effect). From this study, I do not see any clear evidence concerning the importance of the motor stimulation in relation to ASD.

From my perspective, the main contribution of the work is the highlighting of variables that discriminates children with or without ASD, using PCA and HCPC. While I find both PCA and the presented HC informative in several ways, the presentation partly mixes regression/clustering with classification purposes. For example, in the conclusion, the authors argue that the “originality of these statistical methods made possible the discrimination between children with ASD and typically developed children according to their motor difficulties”. It is of course clear from the data that the majority of children could be classified correctly using the motor assessments, but no analysis of the precision of such a classification is presented - and no real classification is performed. From the HC I can see three false positives and two false negatives, producing an average classification performance of about 89%. Furthermore, this “classification” is rendered from the complete dataset, and does not distinguish into a training/test-set, thus making it very difficult to know how these results would generalise to a large group. If the authors want to highlight classification e.g. for screening purposes, which I think is an interesting part of their results, I suggest that a proper classification analysis is introduced.

Specifically concerning the AHC and Fig. 2, the dendrogram would be much more informative if it included labels specifying which variables that is used for which cluster. For example, which variable is used to discriminate on the highest (0.2) hight inb the dendrogram? Also, I find the text in the cluster plot of fig. 2 very difficult to read.

And a detailed note, participants were in the age 4 to 12. Still, the age bands used (as specified on line 131) cover ages 7 to 12. Should we implicitly take it that children younger than 7 were on age band one? I recommend this being spelled out.

As a final note, I think that the authors have made a very interesting study and I encourage further analysis of the data, e.g., linking into better understanding of ASD. The authors could e.g., relate to work to Anzulewicz et al. (2016) and potentially find interesting links.

Refs:

Anzulewicz, A., Sobota, K., & Delafield-Butt, J. T. (2016). Toward the Autism Motor Signature: Gesture patterns during smart tablet gameplay identify children with autism. Scientific Reports, 6(July), 1–13.

Author Response

I miss a proper background section (the paper jumps directly from introduction to methods), but the absence of background is to a large degree compensated by a literature review included in the discussion.

We added a background part in the introduction: Therefore, it seems essential to stimulate children with ASD through a wide range of physical activities. Then, evaluating all motor skills is important in order to properly identify the specific needs of these children.

The main finding highlighted is that flexibility, explosive power, strength, dexterity, and ball skills are lower in children diagnosed with ASD, compared to controls. Most of these results aligned with previous research, and here I would like to see a clearer presentation of what is new.

We added a precision: Our results therefore confirmed the motor deficits in children with ASD. The novelty in this work was the selection of relevant variables that made it possible to properly discriminate between the types of children according to their motor profile. Thus, it allowed us to obtain interesting information concerning the care of a child by evaluating him using the 8 tests selected by the PCA without having to undergo a long evaluation on a large number of tests.

 The authors claim that “These results also confirmed the importance of the motor stimulation at a very early age in order to develop a complete motor profile, adequate skills, and to increase their effectiveness.” In this regard I’m much more sceptical. Firstly, the age span is large, starting at the age of 4, and it’s difficult to assess anything about children in a “very early age” based on this data

I don't quite understand your remark.

We don't have any 4-year-old children in this work. As shown in Table 1, all children are over the age of 8. This is why we only selected MABC evaluations for these age groups (> 8 years)

Secondly, and even more problematic, is that the study demonstrates a correlation between motor skills and ASD (not a casual effect). From this study, I do not see any clear evidence concerning the importance of the motor stimulation in relation to ASD.

We agree with you. Nevertheless, we proposed an evaluation that showed lower results for the ASD group. However, with the ACP and ACHA methods, we can see that three children with ASD (23, 24, 27) very involved in the sport activity obtained very good results.

This is why the physical activity must be encouraged because it will make it possible to limit motor deficits and to promote learning.

From my perspective, the main contribution of the work is the highlighting of variables that discriminates children with or without ASD, using PCA and HCPC. While I find both PCA and the presented HC informative in several ways, the presentation partly mixes regression/clustering with classification purposes. For example, in the conclusion, the authors argue that the “originality of these statistical methods made possible the discrimination between children with ASD and typically developed children according to their motor difficulties”. It is of course clear from the data that the majority of children could be classified correctly using the motor assessments, but no analysis of the precision of such a classification is presented - and no real classification is performed.

Some precisions are added after the dendrogram, and with the ROC curve, added in the document.

Furthermore, this “classification” is rendered from the complete dataset, and does not distinguish into a training/test-set, thus making it very difficult to know how these results would generalise to a large group.

We used the elastic net method that simultaneously permits automatic selection of pertinent variables. It can also select groups of correlated variables.  We did not classify the subjects on all variables but only on the 8 selected. This is corrected in the new document.

Specifically concerning the AHC and Fig. 2, the dendrogram would be much more informative if it included labels specifying which variables that is used for which cluster. For example, which variable is used to discriminate on the highest (0.2) hight inb the dendrogram?

Labels are now added and explained in the legend of the dendogram

Also, I find the text in the cluster plot of fig. 2 very difficult to read.

Yes totally agree. I changed this with a new figure

And a detailed note, participants were in the age 4 to 12.

No, not at all !

Still, the age bands used (as specified on line 131) cover ages 7 to 12. Should we implicitly take it that children younger than 7 were on age band one? I recommend this being spelled out.

This is clearly specified in table 1 (with the min and max values) and in the paragraph (participants): all children are over the age of 8.

As a final note, I think that the authors have made a very interesting study and I encourage further analysis of the data, e.g., linking into better understanding of ASD. The authors could e.g., relate to work to Anzulewicz et al. (2016) and potentially find interesting links.

We read this work with interest but the age range isn’t the same as well as the evaluations, which are quite different in terms of motor skills.